# Binding of Immunoglobulin G to Protoporphyrin IX and Its Derivatives: Evidence the Fab Domain Recognizes the Protoporphyrin Ring

**DOI:** 10.3390/antib8010006

**Published:** 2019-01-04

**Authors:** Koichi Orino

**Affiliations:** Laboratory of Veterinary Biochemistry, School of Veterinary Medicine, Kitasato University, Aomori 034-8628, Japan; orino@vmas.kitasato-u.ac.jp; Tel.: +81-176-23-3471; Fax: +81-176-23-8703

**Keywords:** antibody, Fab domain, Fc domain, protoporphyrin, zinc

## Abstract

Immunoglobulin G (IgG) is known to bind zinc via the Fc domain. In this study, biotinylated protoporphyrin IX (PPIX) was incubated with human IgG and then zinc-immobilized Sepharose beads (Zn-beads) were added to the mixture. After washing the beads, the binding of biotinylated PPIX with IgG trapped on Zn-beads was detected using alkaline phosphatase (ALP)-labeled avidin. Human IgG and its Fab domain coated on microtiter plate wells recognized biotin-labeled PPIX and its derivatives, Fe-PPIX and Zn-PPIX, whereas the Fc domain showed some extent of reaction only with Zn-PPIX. When rabbit anti-bovine transferrin (Tf) antibodies were incubated with biotinylated PPIX, the binding of anti-Tf antibodies with apo-Tf was indirectly detected using ALP-labeled avidin, suggesting that even if the antibody is modified with PPIX, the antibody-antigen reaction occurs. These results suggest that the IgG Fab domain recognizes PPIX and its derivatives, probably via the recognition of the PPIX ring. It is unlikely that binding between the Fab domain and PPIX affects the Fc domain-zinc interaction or antigen-antibody reaction.

## 1. Introduction

In 1890, von Behring and Kitasato reported the presence of an agent in the blood that functioned as an anti-toxin, neutralizing diphtheria toxin, and this agent was eventually identified as the antibodies that control infection in body tissues [1,2]. Over a century later, antibodies are used therapeutically to prevent many types of infectious disease by exploiting their ability to bind a variety of pathogens, including viruses, bacteria, and parasites [1,2,3]. Antibodies are used to treat many types of medical conditions, not only due to their ability to challenge infection as a replacement therapy but also due to their anti-inflammatory and immunomodulatory effects [4]. Antibodies are members of the immunoglobulin (Ig) superfamily that are synthesized in Bursa lymphocytes (B-cells) and constitute up to 20% of total plasma protein [5]. 

Most Igs have common features, including two identical heavy chains and two identical polypeptide light chains, which are bound by disulfide bonds to form a Y-shaped protein [1,2,5]. These macromolecular glycoproteins have a molecular weight greater than 150 kDa [5]. Igs also contain two fragment, antigen-binding (Fab) domains, which recognize antigens with high selectively and specificity, and two fragment, crystallizable (Fc) regions that can bind surface receptors on effector cells, immune proteins, or other antibodies [1,2,5]. 

Gram-positive bacteria produce Ig-binding proteins such as Protein A, G, I, and Z as virulence factors [5]. Because these proteins recognize Fc and/or Fab domains, they can be used in purified form to isolate Igs. IgG can bind zinc via the Fc domain [6]. In humans, the binding of albumin and IgG with protoporphyrin IX (PPIX) had been reported previously [7,8]. In the present study, the mechanism of binding between IgG and PPIX was elucidated. The present data suggest that IgG/PPIX binding could be useful in immunotherapy by exploiting binding between PPIX-drug conjugates and IgG.

## 2. Materials and Methods

### 2.1. Chemicals

Hemin (Fe(Ⅲ)-PPIX chloride), hemin agarose, agarose, and bovine apo-transferrin (Tf) were purchased from Sigma Chemical Co. (St. Louis, MO, USA). PPIX and Zn-PPIX were purchased from Frontier Scientific Inc. (Logan, UT, USA). Human and rat IgGs were purchased from Invitrogen Corp. (Carlsbad, CA, USA). Human IgG Fab and Fc were purchased from Rockland Immunochemicals Inc. (Limerick, PA, USA). Chelating Sepharose^TM^ Fast-Flow and Sepharose 4B were purchased from GE Healthcare (Cleveland, OH, USA). NeutrAvidin alkaline phosphatase (ALP)-conjugated, EX-Link biotin-PEO_2_-amine, 1-ethyl-3-[3-dimethylaminopropyl]carbodiimide hydrochloride (EDC), and Pierce Bicinchochonic acid (BCA) protein assay kit and Immuno Plate Maxisorp F96 microplates were purchased from Thermo Fisher Scientific Inc. (Rockford, IL, USA). Block Ace (BA) was purchased from DS Farma Biomedical Co., Ltd. (Suita-shi, Osaka, Japan). Cellulose ester tubes (Spectra/Por^®^ 6 membrane; molecular weight cut-off [MWCO]: 1,000 Da) and other analytical-grade chemicals were purchased from Wako Pure Chemical Industries, Ltd. (Chuo-ku, Osaka, Japan). Purified water (Elix water) was produced from tap water using an Elix Advantage Water Purification System (Millipore, Billerica, MA, USA).

### 2.2. Antibodies to Bovine Tf

Antiserum to bovine Tf was prepared by immunizing rabbits as described previously [9], and anti-Tf antibodies were purified according to a previously described method [10]. All procedures were performed according to the guidelines of the Animal Care and Use Committee of the School of Veterinary Medicine at Kitasato University (approval no. 17-004).

### 2.3. Preparation of PPIX and PPIX Derivatives

Hemin solution was freshly prepared by dissolving 5 mg of hemin in a minimal amount of 0.1 M NaOH, which was then diluted with phosphate-buffered saline (PBS: 150 mM NaCl, 20 mM sodium phosphate (pH 7.2)) to provide a final concentration of 10 mM NaOH and 1 mg/mL hemin. PPIX and Zn-PPIX were prepared using the same method described above for the preparation of hemin. Concentrations of hemin and Zn-PPIX were determined based on the molar extinction coefficient (ε_385 nm_ = 58,400 M^−1^·cm^−1^ and ε_412 nm_ = 87,400 M^−1^·cm^−1^ for hemin and Zn-PPIX, respectively), as previously described [11,12]. The PPIX concentration was also determined based on its molar extinction coefficient (ε_405 nm_ = 124,000 M^−1^·cm^−1^) after dissolving in dimethyl formamide [13].

### 2.4. Biotinylation of PPIX, Hemin, and Zn-PPIX

A solution of hemin was freshly prepared by dissolving 5 mg of hemin in a minimal amount of 0.1 M NaOH. Hemin was biotinylated with soluble EZ-Link Biotin-PEO_2_-amine at a molar ratio of EZ-Link Biotin-PEO_2_-amine to hemin of 2:1 using EDC as the cross-linking reagent, as previously described [10]. The mixture was then exhaustively dialyzed against PBS using cellulose ester tubes (MWCO: 1000) to obtain biotinylated hemin. Biotinylated PPIX and Zn-PPIX were also prepared as described above. Concentrations of hemin, Zn-PPIX, and PPIX were determined based on their molar extinction coefficients, as described above.

### 2.5. Binding of Zn-Binding IgG to Biotinylated PPIX

Zinc ion was immobilized on chelating Sepharose Fast-Flow beads using 0.2 M ZnSO_4_ according to the manufacturer’s instructions, and the beads were then suspended to 50% (*v*/*v*) in PBS. Next, 1 mL of PBS containing human IgG (25 µg) and 1.5 µM biotinylated PPIX was placed in a 1.5-mL microtube, and the mixture was rotated at 4 °C for 1 h. After the formation of the complex between IgG and biotinylated PPIX, a 50-µL suspension of 50% (*v*/*v*) Zn-beads or Sepharose 4B beads (control beads: CB) was added (net volume of beads per sample: 25 µL), and the mixture was rotated at 4 °C for 1 h. The mixture was then centrifuged at 14,000× *g* for 5 min at 4 °C. The supernatant was discarded, 1 mL of washing solution (0.5 M NaCl, 20 mM sodium phosphate buffer (pH 7.2)) was added to the pelleted beads, and the suspension was centrifuged as described above. Next, 1 mL of washing solution was added, and the beads were suspended and centrifuged again as described above. Finally, 1 mL of PBS was added to the pelleted beads, and the suspended beads were again centrifuged. The resultant pelleted beads were suspended with 1 mL of ALP-labeled avidin (0.5 µg/mL) in PBS and transferred into a Spitz glass tube, which was masked with an appropriate amount of 0.1% (*w*/*v*) BA at room temperature for 1 h. The mixture was incubated at room temperature for 45 min, and the beads were then washed using the same method described above, with the exception that centrifugation was performed at 1700× *g* for 15 min instead of 14,000× *g* for 5 min. After washing, the enzyme reaction was initiated by the addition of 1 mL of 3 mM disodium *p*-nitrophenyl phosphate to the pelleted beads in the tube, and the mixture was centrifuged at 1700× *g* for 5 min after allowing the reaction to occur. The absorbance of the resulting supernatant was measured at 405 nm.

### 2.6. Binding of Human IgG, Fab and Fc to Biotinylated PPIX, Hemin and Zn-PPIX

A total of 100 µL of commercial human IgG, Fab and Fc dissolved in PBS (100 µg/mL) was added to immunoplate wells and incubated overnight at 4 °C. The wells were then washed three times with PBS containing 0.05% Tween 20 (PBST), and the wells were masked with 1% BA for 1 h at room temperature. The wells were then washed, and a 100-µL aliquot of biotinylated hemin (1 µM), PPIX (2 µM), or Zn-PPIX (2 µM) in PBS was added to each well, and the plate was incubated at 37 °C for 1 h. After washing with PBST, 100 µL of ALP-labeled avidin (0.5 µg/mL) in PBS was added to each well for the direct detection of biotinylated PPIX and its derivatives, and the plate was incubated at 37 °C for 45 min. After washing, the enzyme reaction was initiated using 3 mM disodium *p*-nitrophenyl phosphate, and the absorbance of each well was measured at 405 nm using a Molecular Devices VersaMax^TM^ absorbance tunable microplate reader, as previously described [10]. BCA protein assay kit is used for determining the their protein concentrations. 

### 2.7. Effect of the Complex Formed between Antibody and Biotinylated PPIX on the Antibody-Antigen Reaction

A 100-µL volume of a solution of bovine apo-Tf (5 µg/mL) in PBS was added to immunoplate wells and incubated at 4 °C overnight. The wells were then washed three times with PBST and masked with 1% BA for 1 h at room temperature. Next, 1 mL of PBS containing rabbit anti-bovine apo Tf antibodies (100 µg/mL) and biotinylated PPIX (1 µM) was added and incubated at 4 °C for 2 h. The mixture was then dialyzed against 1 L of PBS with traditional dialysis tube. After dialysis, the mixture was centrifuged at 14,000× *g* for 5 min. Biotinylated PPIX (1 µM) or anti-Tf antibodies (100 µg/mL) were treated according to the same method described above. The resultant supernatant (100 µL each) was added to wells coated with apo-Tf, and the plate was incubated at 37 °C for 1 h. After washing with PBST, 100 µL of ALP-labeled avidin (0.5 µg/mL) in PBS was added to each well for the direct detection of biotinylated PPIX, and the plate was incubated at 37 °C for 45 min. After washing, the enzyme reaction was initiated using 3 mM disodium *p*-nitrophenyl phosphate, and the absorbance of each well was measured at 405 nm. Absorbance values were calculated considering non-specific reactions due to non-coating protein.

### 2.8. Statistical Analysis

All data are expressed as the mean ± standard deviation (SD) of four measurements. The Student’s *t*-test was used to compare differences between two groups. Significant differences in multiple comparisons were assessed using one-way analysis of variance followed by Tukey’s test. A *p*-value < 0.01 was considered indicative of a statistically significant difference.

## 3. Results and Discussion

### 3.1. Fab-Mediated Binding of IgG to PPIX and Its Derivatives

Previous research utilizing Zn-beads demonstrated that human IgG binds to zinc ions via the Fc domain but not Fab domain [6]. In this study, IgG-binding biotinylated PPIX also bound to Zn-beads (Figure 1). IgG coated onto microplate wells bound to biotinylated PPIX and its derivatives (hemin and Zn-PPIX), as shown in Figure 2. The IgG Fab domain also bound to them, but the Fc domain exhibited only slight binding to Zn-PPIX. The minimal binding between the Fc domain and Zn-PPIX may be due to the low binding affinity between the Fc domain and zinc ion within the PPIX ring [6], although the contamination of the undigested IgG or Fab fragment in the Fc domain sample may influence this binding. By contrast, the Fab domain likely recognizes the PPIX ring directly. The present results suggest that IgG binds to both PPIX and zinc ion and circulates as a zinc- and/or PPIX-binding protein.

Various gram-positive bacteria produce Ig-binding proteins that recognize the Fc domain, although some of these proteins bind the Fab domain [5]. The binding affinity of these Ig-binding bacterial proteins can be exploited to purify IgG. The results of the present study indicate the possibility that PPIX and its derivatives could also be utilized for the purification of IgG.

### 3.2. Effect of Antibody-PPIX Interactions on Antibody-Antigen Reactions

From the perspective of Fc-binding proteins, peptide ligands that reportedly bind to the Fc domain of IgG are attractive immunotherapy candidates [5]. As the Fab domain was found to bind to PPIX, the effect of the interaction between IgG and PPIX on antibody-antigen reactions was also examined (Figure 3). Interestingly, ALP-conjugated anti-bovine Tf antibodies also functioned as heme-binding proteins, as the ALP-labeled antibodies exhibited significantly greater binding to hemin-agarose than agarose alone (Appendix A). After the incubation of the rabbit anti-bovine Tf antibody with biotinylated PPIX, the mixture was dialyzed against PBS to remove free biotinylated PPIX, and the dialyzed material was then added to microplate wells coated with bovine apo-Tf. Complexes formed between the anti-Tf antibody and biotinylated PPIX were detected using ALP-labeled avidin. These results suggest that biotinylated PPIX does not completely interfere with antibody-antigen reactions, and thus there is a possibility facilitating the development of noncovalent antibody-drug conjugates with cross-linked PPIX and drug. Natural PPIX may not elicit the production of natural antibodies directed against itself, in contrast to artificial peptide ligands. In addition, the use of a monospecific monoclonal antibody may give technological advancement as in NISTmAb reference material 8671 [14]. PPIX is stable in 10% FBS supplemented medium for 4 h or more [15], suggesting that PPIX is physically stable in blood. However, further study needs to clarify the effect of PPIX-binding IgG on the antibody-antigen reaction due to the different affinities of antibodies to their respective antigen. In humans, fibrinogen and albumin are also known as a PPIX and/or heme-binding protein in addition to IgG [16], and these proteins may play a protective role against oxidative stress caused by PPIX itself and/or iron as radical scavenger [15,16]. On the other hand, albumin shows stronger binding to PPIX than IgG [8]. It remains to clarify the competitive binding of PPIX-binding IgG by PPIX and/or other PPIX-binding proteins in the circulation. As described above, different IgG domains are involved in the binding of PPIX and zinc ions. Zinc is considered a ‘miracle’ element due to its anti-oxidant and anti-inflammatory properties [17], which provide additional benefits that could be obtained from the exploitation of Zn-binding IgG [6]. It is noteworthy that immunotherapies exploiting the binding of IgG to PPIX could be accompanied by zinc-associated beneficial effects.

## 4. Conclusions

The complex formed between human IgG and biotinylated protoporphyrin IX (PPIX) still showed the binding with zinc-immobilized Sepharose beads (Zn-beads) via the Fc domain. Human IgG and its Fab domain recognized biotin-labeled PPIX and its derivatives, Fe-PPIX and Zn-PPIX, whereas the Fc domain showed some extent of reaction only with Zn-PPIX. After the incubation of the rabbit anti-bovine transferrin antibodies with biotinylated PPIX, the binding of anti-Tf antibodies with apo-Tf was indirectly detected using ALP-labeled avidin, suggesting that the antibody modified with PPIX can bind antigen. These results suggest that the IgG Fab domain recognizes PPIX and its derivatives, and that binding between the Fab domain and PPIX seems not to affect the Fc domain-zinc interaction or antigen-antibody reaction.

## Figures and Tables

**Figure 1 antibodies-08-00006-f001:**
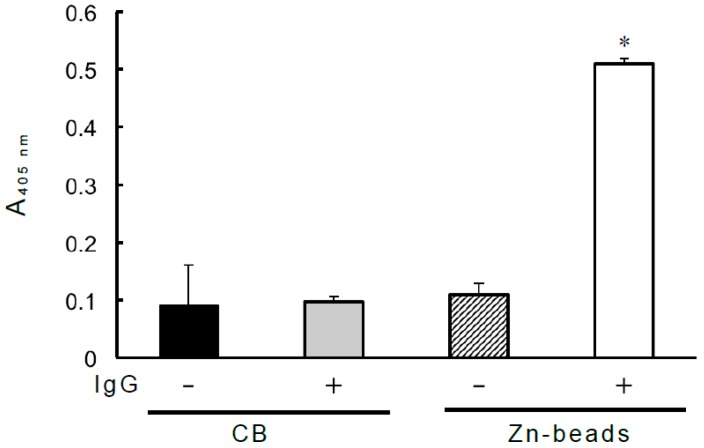
Zn-binding IgG binds to biotinylated PPIX. A 1-mL volume of PBS containing human IgG (25 µg) and 1.5 µM biotinylated PPIX was added to a 1.5-mL microtube, and the mixture was rotated at 4 °C for 1 h. After incubation, a suspension of 50 µL of 50% (*v*/*v*) Zn-beads or control beads (CB) was added (net volume of beads per sample: 20 µL) to the microtube, and the mixture was rotated at 4 °C for 1 h. The mixture was then centrifuged at 14,000× *g* for 5 min at 4 °C, and the pelleted beads were washed three times as described in the “Materials and methods.” Finally, 1 mL of ALP-labeled avidin (0.5 µg/mL) in PBS was added, and the suspension was transferred to a Spitz glass tube masked with 0.1% (*w*/*v*) BA. The mixture was incubated at room temperature for 45 min. After incubation, the beads were washed as described in the “Materials and methods.” After washing, the ALP reaction was carried out with 3 mM disodium *p*-nitrophenyl phosphate, and the release of *p*-nitrophenol was monitored spectrophotometrically at 405 nm. Each value is the mean ± SD of triplicate determinations. * *p* < 0.01 versus binding examined with CB in the absence of IgG (solid column).

**Figure 2 antibodies-08-00006-f002:**
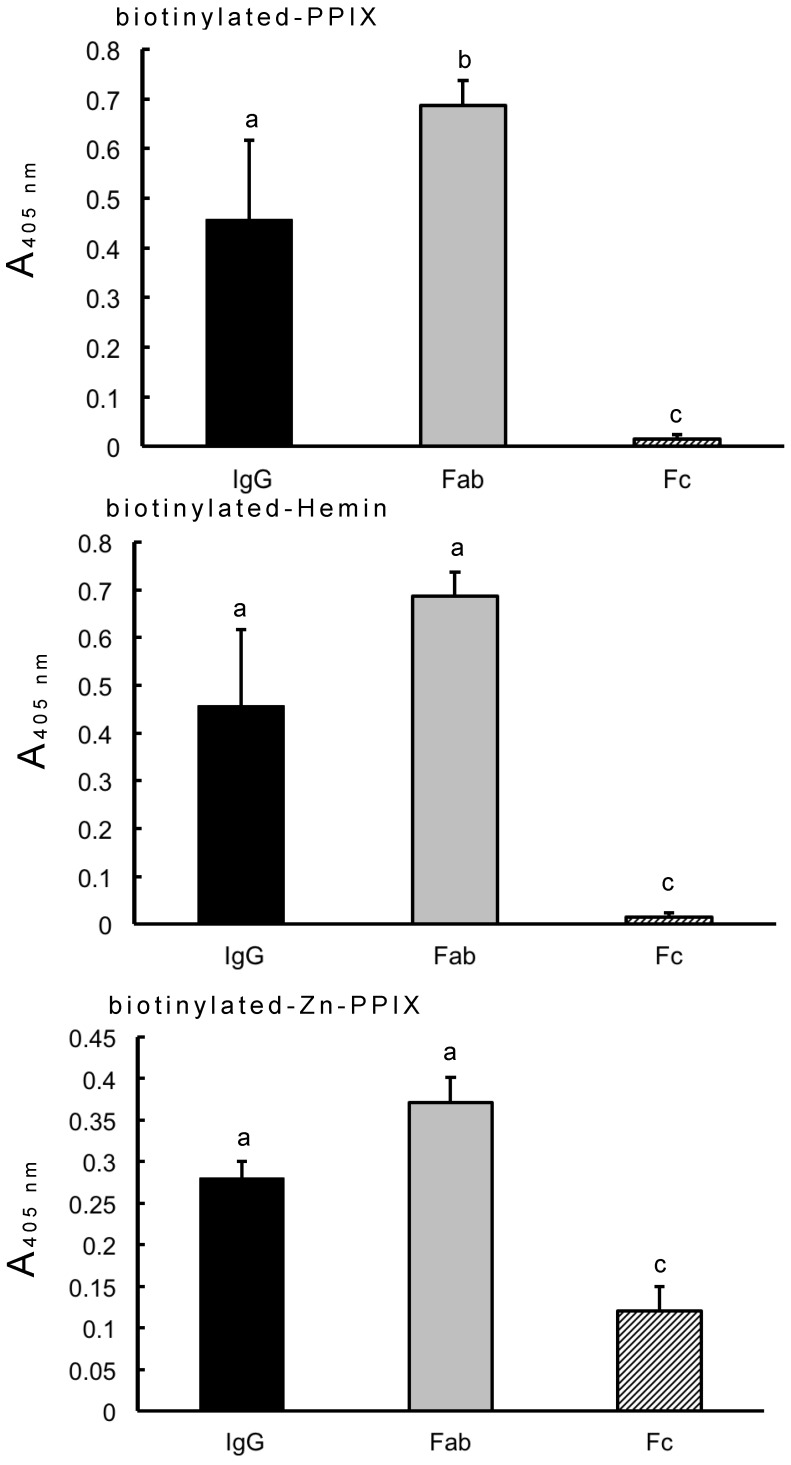
Binding of human IgG and IgG Fab and Fc domains to biotinylated PPIX, hemin, and Zn-PPIX. Human IgG and its Fab and Fc domains were dissolved and coated on microplate wells (500 ng/well), and then a 100-µL aliquot of biotinylated hemin (1 µM), PPIX (2 µM), or Zn-PPIX (2 µM) in PBS was added to each well, and the plate was incubated at 37 °C for 1 h. After washing with PBST, 100 µL of ALP-labeled avidin (0.5 µg/mL) in PBS was added to each well, and the plate was incubated at 37 °C for 45 min. After washing, the ALP reaction was carried out as described in Figure 1. Each value is the mean ± SD of four replicates. a, b, c: identical letters indicate no significant difference; a versus b: *p* < 0.05; b versus c: *p* < 0.01; a versus c: *p* < 0.01.

**Figure 3 antibodies-08-00006-f003:**
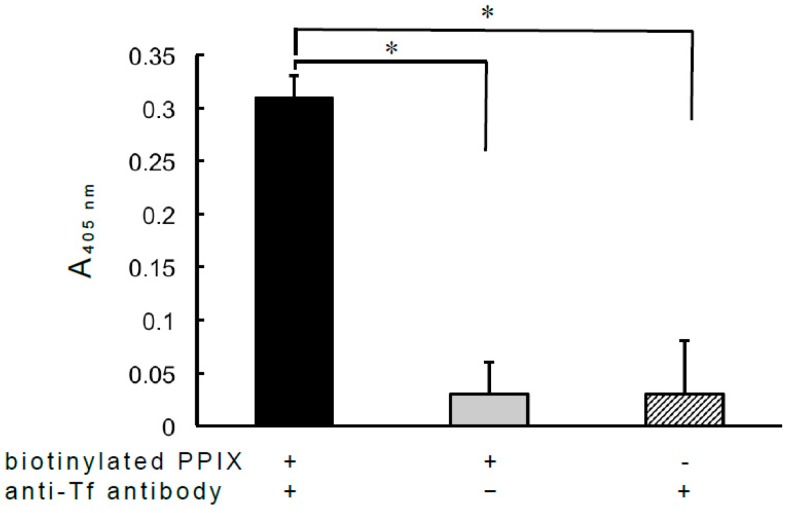
Effect of the antibody–biotinylated PPIX complex on antibody-antigen reactions. A total of 1 mL of PBS containing rabbit anti-Tf antibodies (100 µg/mL) and biotinylated PPIX (1 µM) was incubated at 4 °C for 2 h, after which the mixture was dialyzed against PBS. After dialysis, the mixture was centrifuged at 14,000× *g* for 5 min, and the resultant supernatant (100 µL) was added to a microplate well coated with bovine apo-Tf (500 ng/well), as described in “Materials and methods,” and the plate was then incubated at 37 °C for 1 h. After washing, 100 µL of ALP-labeled avidin (0.5 µg/mL) in PBS was added to each well for the direct detection of biotinylated PPIX, and the plate was incubated at 37 °C for 45 min. After washing, the enzyme reaction was performed as already described. Aliquots (100 µL) of biotinylated PPIX (1 µM) or anti-Tf antibodies (100 µg/mL) were also added to microplate wells coated with apo-Tf, and the plate was treated as described above. Each value is the mean ± SD of three replicates, and the absorbance of each well was measured at 405 nm as described above. Each absorbance value was calculated considering nonspecific reactions associated with non-coating protein. * *p* < 0.01 versus binding examined with Tf in the absence of anti-Tf antibodies or biotinylated PPIX.

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
