# Peer review of "Binding of Immunoglobulin G to Protoporphyrin IX and Its Derivatives: Evidence the Fab Domain Recognizes the Protoporphyrin Ring"

_2073-4468, 2019, doi:10.3390/antib8010006_

Round 1

Reviewer 1 Report

Introduction

Lines 25-45. 

Description about antibody should be shortened. The author should describe biological and biochemical profiles of protoporphyrin IX in more detail.

Lines 45-47.  “The author found that human IgG isolated using Sepharose beads with immobilized zinc ions (Zn-beads) recognizes protoporphyrin IX (Supplementary Data, Fig. S1).”  

This sentence should be omitted from Introduction.

Results and Discussion

Lines 144-145. 

The sentence “The present demonstrated the binding of Zn-binding IgG to PPIX basedon the lower recovery of PPIX as compared with agarose beads (Supplementary Data, Fig. S1).” should be corrected for instance as follows; “The fact that recovery amount of PPIX from Zinc ion-chelated Sepharose beads was lower than that from control beads in the presence of immunoglobulin (Supplementary Data, Fig. S1) suggested that PPIX bound to IgG.”  However, Fig. S1 can be omitted because it overlaps the data shown in Fig. 1.

Lines 191-193. 

If the author wants to describe that biotinylated PPIX does not interfere antibody-antigen reactions, the affinities of some antibodies against their respective antigens should be compared in the presence and absence of PPIX.  It cannot be concluded that “PPIX does not interfere antibody-antigen reaction” from the results in this paper.

Moreover, if the author proposes ADC utilizing PPIX, the author should describe in Discussion not only the binding stability between antibody and PPIX, but also physical stability of PPIX in blood.

Especially, the author should consider the facts that PPIXbound to human serum albumin more than three times higher than to IgG (Biophysic. Chem., 109: 351-360, 2004) and that binding constants to albumins and the immunoglobulin were comparable (Biophysic. Chem., 96: 77-87, 2002).

Author Response

To Reviewers 1 and 2

    I am very thankful for view letter of Dec 5, 2018, with valuable comments. I am sending my manuscript (Antibodies-398773) with major revisions as suggested by Reviewers 1 and 2:

1)   Supplementary data S1: this data had been deleted as suggested by Reviewer 1. Descriptions on Figure S1 and S2 had also been removed or changed (Line 182; Lines 229-130).

2)   We had inserted four manuscripts in “References”. Reference Nos. 7 and 8 had been used for the binding of human serum albumin and IgG with PPIX (also see Lines 42-44). No. 15 had been added for explanation of stability of PPIX in blood (also see Lines 191-192). In addition, Nos. 7 and 16 had been inserted as data of other PPIX-binding proteins as in albumin and fibrinogen, suggesting that these proteins compete the binding of IgG with PPIX (also see Lines 192-198). In addition, the manuscripts (Nos. 15 and 16) were used for physiological role for protective role against oxidative stress by light or iron by these proteins (also see Lines 195-197). 

To Reviewer 1

1)   The sentence has been inserted for application of ADC as described above (Lines 197-198).

    I hope that all these corrections and revisions will be satisfactory, and that the revised version will be acceptable for publication in Antibodies.

Sincerely yours

Koichi Orino

Reviewer 2 Report

Review of manuscript submitted by Orino to Antibodies: Binding of immunoglobulin G to protoporphyrin IX and its derivatives: Evidence the Fab domain recognizes the protoporphyrin ring (antibodies-398773).

The manuscript demonstrates that the Fab domain is involved in the binding of protoporphyrin IX and its derivatives to IgGs, and that the binding does not seem affect antigen-antibody binding.

The experiments are well performed and the manuscript well written. It would have been interesting to see affinity data included, also for human IgG1, IgG2 etc., F(ab')2 of human IgG1-4 could be generated using the FabRICATOR (IdeS) protease.

Depending on the origin of the Fc domain (recombinant or purified after IgG endoprotease digest), I suggest to mention that the minimal binding to Fc domain might also be influenced by contaminated FAP or undigested IgG, or incompletely digested IgG with one Fab domain.

Potential in vivo implications of the IgG circulating as a zinc- and/or PPIX-binding protein should be discussed.

Minors:

1) Line 40: Modify ‘two fragment, crystallizable (Fc) regions’ to ‘one fragment, crystallizable (Fc) region’

2) Line 42: Change ‘ternover’ to ‘turnover’ (??)

3) Figure 1 and S1: Please add ‘control beads’ for CB in the legend

4) I recommend to shorten all figure legends and not repeat sections from Materials and Methods. Otherwise hard to read.

Author Response

To Reviewers 1 and 2

    I am very thankful for view letter of Dec 5, 2018, with valuable comments. I am sending my manuscript (Antibodies-398773) with major revisions as suggested by Reviewers 1 and 2:

1)   Supplementary data S1: this data had been deleted as suggested by Reviewer 1. Descriptions on Figure S1 and S2 had also been removed or changed (Line 182; Lines 229-130).

2)   We had inserted four manuscripts in “References”. Reference Nos. 7 and 8 had been used for the binding of human serum albumin and IgG with PPIX (also see Lines 42-44). No. 15 had been added for explanation of stability of PPIX in blood (also see Lines 191-192). In addition, Nos. 7 and 16 had been inserted as data of other PPIX-binding proteins as in albumin and fibrinogen, suggesting that these proteins compete the binding of IgG with PPIX (also see Lines 192-198). In addition, the manuscripts (Nos. 15 and 16) were used for physiological role for protective role against oxidative stress by light or iron by these proteins (also see Lines 195-197). 

To Reviewer 2

1)   The sentence has been added for minimal binding of Fc domain (Line 146).

2)   The explanation of CB had been inserted as “control beads” (Lines 156-157).

    I hope that all these corrections and revisions will be satisfactory, and that the revised version will be acceptable for publication in Antibodies.

Sincerely yours

Koichi Orino

Round 2

Reviewer 1 Report

Lines 143-145: “The minimal binding between the Fc domain and Zn-PPIXmay be due to low binding affinity between the Fc domain and zinc ion within the PPIX ring [6],although contamination of undigested IgG or Fab fragment may influence this binding.”

Referee’s comment: If undigested IgG or Fab fragment was contaminated in the Fc domain sample, similar binding must be observed even with biotinylated PPIX. Therefore, the author does not have to add the following sentence “although contamination of undigested IgG or Fab fragment may influence this binding”.

Lines 150-151: “The results of the present study indicate that PPIX and its derivatives could also be utilized for the purification of IgG.”

Referee’s comment: If the author wants to describe the possibility of utility of PPIX for IgG purification, the author should consider the binding specificity and affinity of PPIX to IgG in comparison with some other proteins referring articles or the author’s own data. 

Lines 179-181: “Interestingly, ALP-conjugated anti-bovine Tf antibodies also functioned as heme-binding proteins, as the ALP-labeled antibodies exhibited significantly greater binding tohemin-agarose than agarose alone (Supplementary Data, Fig. S1).”

Referee’s comment: It seems that Supplementary has not been revised. Fig. S1 should be omitted from it.

Lines 185-187 and 191-193: “These results suggest that biotinylated PPIX does not interferewith antibody-antigen reactions, thus facilitating the development of noncovalent antibody-drug conjugates with cross-linked PPIX and drug. “ and “However, further study needs to clarify the effect of PPIX-binding IgG on antibody-antigen reaction due to different affinities of antibodies to their respective antigen. “

Referee’s comment:The author presented the effect of the antibody–biotinylated PPIX complex on antibody-antigen reactions in Fig.3. However, these results do not show how much PPIX affects the binding affinity of the antibody to its antigen. The author should compare binding affinity of the anti-Tf antibody to its antigen at least in the presence and absence of PPIX by surface plasmon resonance or ELISA.

Author Response

To Reviewers 1

    I am very thankful for view letter of Dec 12, 2018, with valuable comments. I am sending my manuscript (Antibodies-398773) with major revisions as suggested by Reviewers 1:

1)   Lines 146-147: The following sentence had been inserted as suggested Reviewer 1; “although contamination of undigested IgG or Fab fragment in FC domain sample may influence this binding”.

2)   Lines 152-153: We had inserted “the possibility” as suggested by Reviewer 1 (Lines 152-153), although further study needs to examine applicability of PPIX and its derivatives to purify IgG.

3)   We had two Supplemental Data, and we had deleted Fig. S1 as suggested by Reviewer 1. Eventually, S2 had been changed into “S1”. I am sorry for this inconvenience.

4)   Lines 185-187: Firstly, anti-Tf antibody was incubated with biotinylated PPIX, and the unbound PPIX was removed by dialysis. After removal of the unbound PPIX, the complex of anti-Tf antibody with biotinylated PPIX bound plate-coated Tf with the detection of ALP-avidin. Although this ELISA system demonstrated the possibility of ADC, further study needs the effect of PPIX and other PPIX-binding in blood on this ADC method as suggested by Reviewer 1. Finally, the sentence (Line 199) had been changed. 

    I hope that all these corrections and revisions will be satisfactory, and that the revised version will be acceptable for publication in Antibodies.

Sincerely yours

Koichi Orino

Round 3

Reviewer 1 Report

If you do not add any results which I suggested, you should change the sentences 185-187 to "These results suggest that biotinylated PPIX does not completely interfere with antibody-antigen reactions." 

I do not think that this result will leads to "facilitating the development" . Therefore, you should describe it such as "there is a possibility..".

Author Response

To Reviewers 1

    I am very thankful for view letter of Dec 18, 2018, with valuable comments. I am sending my manuscript (Antibodies-398773) with minor revisions as suggested by Reviewers 1:

1)   Lines 187-189: The sentence had been changed as suggested Reviewer 1 as follows: “These results suggest that biotinylated PPIX does not completely interfere with antibody-antigen reactions, and thus there is a possibility facilitating the development of noncovalent antibody-drug conjugates with cross-linked PPIX and drug.”.

    I hope that all these corrections and revisions will be satisfactory, and that the revised version will be acceptable for publication in Antibodies.

Sincerely yours

Koichi Orino
